# Processing Optimization of Shear Thickening Fluid Assisted Micro-Ultrasonic Machining Method for Hemispherical Mold Based on Integrated CatBoost-GA Model

**DOI:** 10.3390/ma16072683

**Published:** 2023-03-28

**Authors:** Jiateng Yin, Jun Zhao, Fengqi Song, Xinqiang Xu, Yeshen Lan

**Affiliations:** 1College of Mechanical Engineering, Zhejiang University of Technology, Hangzhou 310023, China; 2Key Laboratory of Special Purpose Equipment and Advanced Processing Technology, Zhejiang University of Technology, Ministry of Education & Zhejiang Province, Hangzhou 310023, China; 3School of Mechatronics Engineering, Quzhou College of Technology, Quzhou 324000, China

**Keywords:** surface roughness, micro-hemisphere concave mold (CM), micro-ultrasonic machining (MUM), shear thickening fluid (STF), Categorical Boosting (CatBoost), genetic algorithm (GA)

## Abstract

Micro-electro-mechanical systems (MEMS) hemispherical resonant gyroscopes are used in a wide range of applications in defense technology, electronics, aerospace, etc. The surface roughness of the silicon micro-hemisphere concave molds (CMs) inside the MEMS hemispherical resonant gyroscope is the main factor affecting the performance of the gyroscope. Therefore, a new method for reducing the surface roughness of the micro-CM needs to be developed. Micro-ultrasonic machining (MUM) has proven to be an excellent method for machining micro-CMs; shear thickening fluids (STFs) have also been used in the ultra-precision polishing field due to their perfect processing performance. Ultimately, an STF-MUM polishing method that combines STF with MUM is proposed to improve the surface roughness of the micro-CM. In order to achieve the excellent processing performance of the new technology, a Categorical Boosting (CatBoost)-genetic algorithm (GA) optimization model was developed to optimize the processing parameters. The results of optimizing the processing parameters via the CatBoost-GA model were verified by five groups of independent repeated experiments. The maximum absolute error of CatBoost-GA is 7.21%, the average absolute error is 4.69%, and the minimum surface roughness is reduced by 28.72% compared to the minimum value of the experimental results without optimization.

## 1. Introduction

Compared to traditional gyroscopes, the MEMS hemispherical resonant gyroscope has many advantages, such as high precision, low cost, light weight, and small size [1]. Therefore, it is widely used in electronic products, the aerospace industry, defense industry weapons, industrial robots, etc. [2,3,4]. The hemispherical resonator is located inside the gyroscope and its accuracy directly determines the performance of the gyroscope [5]. The accuracy inside the hemispherical resonator depends mainly on the surface roughness of the micro-hemispherical CM on the silicon wafer surface. Different machining methods for different materials may produce entirely different results, resulting in entirely different surface roughness, machining efficiency, material removal rates, and other parameters [6,7,8,9]. Currently, there are many methods to machine micro-hemispherical concave molds, such as ultrasonic machining and micro-milling. Microelectronic electrical discharge machining of micro-concave molds is efficient, but the edges of the machined micro-concave molds are severely broken and the shape accuracy is low [10]. Micro molds machined by micro-milling have good surface quality but low machining efficiency [11]. Therefore, it is necessary to choose an efficient and effective machining method to process micro-hemispherical CM so as to quickly obtain the micro-concave molds with low surface roughness.

Silicon wafers are hard and brittle materials. Ultrasonic processing has proven to be an excellent method for processing hard and brittle materials. As a result, ultrasonic processing is widely used for the processing of microstructures of various hard and brittle materials with good results. For example, Zhao et al. used ultrasonic processing methods to process silicon wafer microchannels, and the integrity of silicon wafer microchannels was improved by more than 60% [12]. Chen et al. combined electrochemical discharge processing with ultrasonic waves for processing the microstructure of glass plates, resulting in a gap reduction of more than 20% [13]. Zhao et al. used ultrasonic-assisted machining when milling and machining the microstructures on the surface of aluminum plates, and this machining method significantly improved the machining quality of the surface of aluminum plates [14]. Wang et al. greatly improved the shape accuracy of the workpiece surface by filing with ultrasonic processing of carbon fiber reinforced silicon carbide composites [15]. The micro-ultrasonic-assisted grinding process method has proven to be an effective method for processing hard and brittle materials. During the micro-ultrasonic-assisted grinding process, the liquid and abrasive particles within the polishing fluid will fill the space between the tool and the workpiece. Under the vibration of ultrasonic waves, the free abrasive particles will impact the surface of the material and complete the processing of the material surface [16]. Good machining results and high machining efficiency can be obtained by ultrasonic-assisted grinding methods [17].

The composition of the polishing fluid is a key factor in the quality of the hemispherical CM during micro-ultrasonic machining. Therefore, a suitable polishing fluid needs to be used in the ultrasonic processing of hemispherical CM. STF has been used in the ultra-precision polishing field due to its perfect processing performance. Li et al. used the shear-thickening polishing method to polish the surface of mold steel, and the surface roughness of the mold steel was significantly reduced [18]. Li et al. developed an integrated surface roughness model to predict the average surface roughness of shear-thickened polishing with good results [19]. As a result, the application of STF in the MUM of hemispherical CM may result in excellent processing.

Although the use of STF in the MUM of hemispherical CM may give good results, there are still difficulties in predicting and further improving the surface roughness of micro-CM. Therefore, an objective optimization method is needed to predict and optimize the surface roughness of micro-CM during machining. Regression models in machine learning can be trained on a dataset to build a complete mathematical model to achieve good prediction results [20,21,22,23]. Because of the good results of machine learning in model training, it has been widely used in process parameter optimization with good results. For example, Mahjoubi et al. used the tree-based pipeline optimization tool to predict the mechanical properties and economics of strain-hardened cementitious composites, and the predicted results were in excellent agreement with the actual results [24]. Chaki used artificial neural networks with non-dominated ranking GA (NSGA-II) in the multi-objective optimization of the machining quality of laser-cut aluminum alloys, and the accuracy of the optimization results reached more than 99% [25]. Lu et al. employed machine learning to improve the performance of a battery removal platform with a prediction error of less than 10% [26]. Jin et al. applied machine learning to the fabrication of deposited thin film layers in semiconductors and showed that the uniformity of the thickness of the deposited thin film layers was well improved [27]. Zhang et al. applied machine learning methods in the optimization of process parameters for laser-induced plasma micro-machining, with significant improvements in the machining quality of the material [28]. The genetic algorithm is a very effective method for finding optimal solutions and it is used to find optimal solutions [29,30,31]. Zhao et al. employed NSGA-II to the multi-objective optimization of machining efficiency and energy consumption in CNC milling, which resulted in a reduction of energy consumption by more than eight percent and a significant improvement in machining efficiency [32]. Tian et al. used a genetic algorithm to optimize the deformation of T-joint fillet welds [33]. The results showed that the weld deformation became smaller and the deformation differed very little from the model predictions after optimization by the genetic algorithm. Pashazadeh et al. performed process parameter optimization for contact spot welding by means of a genetic algorithm to finalize the tip correction operating range for the welding operation [34]. Alvarado-Iniesta et al. used genetic algorithms to solve the engine mounting problem. Machine learning models and genetic algorithms have already yielded many results in the field of process parameter optimization [35]. CatBoost is a new machine learning model [36,37,38], which, together with genetic algorithms, may be useful in optimizing process parameters for the STF-MUM processing of hemispherical CM for silicon wafers.

In this work, silicon wafer hemispherical CM was processed by STF assisted MUM. Then, a new hybrid CatBoost-GA prediction model was developed for target prediction and the optimization of surface roughness for micro-hemispherical CM based on the surface roughness dataset of hemispherical molds obtained from the processing experiments, in combination with a CatBoost prediction model and GA. The subsequent structure of this paper is as follows. In Section 2, the experimental equipment, experimental design, and experimental results of STF assisted MUM processing of micro-spherical CM are presented. Section 3 describes the principles and the algorithmic structure and process of the new hybrid CatBoost-GA multi-objective optimization model. In Section 4, the performance of CatBoost machine learning and the effect of various process parameters on surface roughness are discussed. In Section 5, the optimal solution of the CatBoost-GA model prediction is experimentally verified. In Section 6, the prediction accuracy and optimization effectiveness of the CatBoost-GA multi-objective optimization model are summarized. The conclusion section summarizes the results of the study.

## 2. Processing Experiment

### 2.1. Experimental Procedure

The overall experimental equipment is shown in Figure 1a. On the left of the experimental equipment, a computer control system and a pressure regulator are included. The overall machining system is on the right of Figure 1a: the overall motion control of the equipment using macro motion control and micro motion control. Macro motion control can achieve rapid positioning of the tool in the XYZ three movement axis, with repeat positioning accuracy of 5 μm. The Z micro-axis is shown in Figure 1b; the linear motor and cylinder are above the Z micro-axis with a positioning accuracy of 100 nm. The air pressure output from the cylinder ensures that the Z micro-axis is balanced prior to machining. The machining tool head is under the cylinder and the tool head is fitted with a hemispherical ceramic tool, which is shown in Figure 1c. The computer control system can regulate the movement of the Z micro-axis with a speed control accuracy of 0.1 μm/s. The ultrasonic processing system is mounted on the Z micro-axis. The ultrasonic transmitter has a power of 120 w and a vibration frequency of 30 kHz, and the power percentage can be adjusted.

The schematic diagram for the STF-MUM processing of a hemispherical CM for silicon wafers is shown in Figure 2a. Prior to processing, the polyhydroxy polymer (PPM) and abrasive particles are first mixed with water in the prescribed proportions to form an STF with abrasive particles. The diagram of the microscopic material removal action during processing is shown in Figure 2b. The abrasive material and STF adhere to the surface of the hemispherical ceramic tool. When ultrasonic vibration occurs, the STF undergoes a shear-thickening effect and solid particles will adhere to the abrasive surface, completing the removal of material from the silicon wafer surface by ultrasonic vibration. It has been demonstrated that tungsten carbide whole ball ceramic tools and zirconia abrasives achieve good results in micro-ultrasonic machining processes. Therefore, in this study, a tungsten carbide spherical ceramic tool and a zirconia abrasive were used; the diameter of the ceramic ball tool was 0.8 mm and the total depth of descent of the tool was set at 300 μm.

### 2.2. Experimental Design

In STF-MUM processing, different concentrations of PPM can result in different viscosity of the solution, which can lead to significant changes in the surface density of the abrasive particles between the tool and the workpiece. In ultrasonic machining, the impact of abrasive particles on the workpiece surface, the surface density of abrasive particles between the tool and the workpiece, and the average particle size of the abrasive particles all affect the material removal from the workpiece surface [39]. The rate of tool descent can lead to differences in machining time, which can also lead to changes in the surface roughness of the workpiece. Therefore, in this study, the concentration of PPM (A), abrasive concentration (B), ultrasonic energy (C), tool lowering speed (D), and the average particle size (E) were the main influencing factors in the experiment, and the output of the experiment was the surface roughness of the micro-CM. In this study, the box model was used for the experimental design, as the box model in the response surface methodology can obtain better analysis with fewer experiments. The parameter levels of the experimental design are shown in Table 1. A total of 5 main factors were considered in the experimental design, while 3 levels and 6 center points were set, and 46 sets of experiments were eventually required.

### 2.3. Experimental Results

The surface roughness of the machined miniature hemispherical CM was measured using a Vecco optical profiler. In measuring the surface roughness of the micro-concave molds, the further away from the center of the circle, the lower the surface roughness and the less obvious the display of surface roughness. Therefore, in this study, with the center of the circle as the central point, the sampling area is 63 μm long and 47 μm wide on the expansion plane, and the surface roughness is expressed as the arithmetic mean (Ra) of the surface roughness.

All experimental results are shown in Table 2.

## 3. Principles of Integrated Algorithms

### 3.1. CatBoost Prediction Model

In order to refine the complete mathematical model with experimental data, this study uses the CatBoost method to train the experimental data. CatBoost is a new gradient boosted decision tree (GBDT) algorithm [40], and it has made many improvements compared to the traditional algorithm.
CatBoost trains the entire dataset and processes the classification features with minimal information loss by means of target statistics.When CatBoost runs, the various different features are combined into one feature. The tree structure in CatBoost is split into individual sub-trees, and combinations are not considered during the first split of the tree. During subsequent splits, CatBoost combines all combinations with all categorical features in the dataset. All splits selected by CatBoost in the tree are considered as two different categories and are used for the combination.A new method to overcome gradient bias is used in CatBoost, which is an ordered boosting method.The forgotten tree data structure is used as a predictor in CatBoost, and the same criteria are used in the hierarchical process of the tree. The forgetting tree is a balanced tree that is not prone to overfitting. Better results can be achieved with these improvements to CatBoost.

In this study, the input vectors and output vectors are shown below:(1)$$X=\left[A,B,C,D,E\right]\;Y=[Ra]$$

Although CatBoost can train the entire dataset, dividing the training set into a test set and a training set can better accomplish the validation effect. Therefore, 37 of the 46 *X*, *Y* datasets were used as the training set and 9 were used as the test set. To test the predictive performance of CatBoost, the obtained CatBoost outputs were compared with the corresponding known experimental outputs and the corresponding results were obtained according to the corresponding evaluation formulae. In order to test the predictive performance of CatBoost, the predicted results of CatBoost were compared with the known experimental output, and the predictive performance of CatBoost was evaluated by the determination coefficient (*R*^2^) and mean square error (*MSE*), as shown in the following equation:(2)$$R_{}^{2}=1-\frac{\sum_{i=1}^{m}{\left(Y_{ci}-\bar{Y}\right)}^{2}}{\sum_{i=1}^{m}{\left(Y_{ci}-Y_{i}\right)}^{2}}$$
(3)$$MSE=\frac{1}{n}\sum_{i=1}^{n}{\left(Y_{ci}-Y_{i}\right)}^{2}$$
where *n* represents the number of samples, *Y_ci_* represents each actual value, *Y_i_* represents each predicted value, and *Y* represents the mean of the samples. In evaluating the performance of machine learning, the closer *R*^2^ is to 1, the better the prediction performance of the machine learning model. When evaluating the performance of machine learning, the smaller the *MSE*, the higher the prediction accuracy of the machine learning model and the better the fit between the prediction results of the machine learning and the experimental results. In the Analysis and Discussion section, the prediction results of CatBoost for the initial experimental dataset are compared with the experimental results to derive the *R*^2^ and *MSE* results and to evaluate the prediction performance of CatBoost.

### 3.2. Multi-Objective Optimization Process

In the optimization of the process parameters for STF-MUM of hemispherical CM, the smaller the Ra of the micro-CM, the better the machining effect; therefore, the optimization equation is defined as follows:

Minimize: ΔR(A)(B)(C)(D)(E).

Based on the accuracy of the experimental apparatus and the analysis of the experimental results, the range of input parameters was determined as follows:(4)0 ≤ A ≤ 35; 1 ≤ B ≤ 15; 60 ≤ C ≤ 120; 0.1 ≤ D ≤ 3; 1 ≤ E ≤ 3
where *A*, *B*, *C*, *D*, and *E* are the input parameters in the process of process parameter optimization and Ra is the process parameter output result, where a smaller Ra means better machining quality.

After the CatBoost model has been trained, a genetic algorithm is used to optimize the surface roughness of the micro-CM. The genetic algorithm initializes the population, calculates the fitness value of the population, and eliminates some of the population. The remaining populations generate new populations by crossover mutation and other operations. The genetic algorithm repeats the previous process until the algorithm stops when the termination condition is reached, at which point the optimal solution obtained in the genetic algorithm is output. The genetic algorithm calculates fitness values with the help of a CatBoost trained model to generate new populations, and the overall process is shown in Figure 3.

The CatBoost-GA model was used to perform the optimization of the process parameters in this study. The model flow chart is shown in Figure 4. CatBoost builds the final prediction model by constructing multiple T CART trees and importing them into the genetic algorithm. The genetic algorithm keeps calculating fitness values from the prediction model in CatBoost and generates new individuals through crossover variation and other operations, ending the operation when the genetic algorithm reaches the number of iterations and produces the optimal solution.

## 4. Analysis and Discussion

### 4.1. Analysis of Variance

In the experiment of STF-MUM hemispherical CM, the surface roughness results of the micro-concave dies were influenced by the combined effect of several factors, so the effect of each factor on the surface roughness of the micro-CM can be analyzed using ANOVA at a significance level of 0.05. The results are shown in Table 3, which indicate that the content of PPM, abrasive particle size, content of PPM × abrasive concentration, content of PPM × ultrasonic energy, content of PPM × average abrasive grain size, abrasive concentration × average abrasive grain size, content of PPM × content of PPM, abrasive concentration × abrasive concentration, ultrasonic energy × ultrasonic energy, abrasive grain size × abrasive grain size were less than 0.05, so these items were the main factors affecting the surface roughness, and, according to the size of the F-value, it can be concluded that the average particle size × the average particle size is the largest influence term, and at the same time the content of PPM in these influence terms is also a very important factor, which shows that the shear thickening fluid plays a great role in the ultrasonic processing.

### 4.2. Performance Evaluation of CatBoost

After the training of the model by machine learning is completed, it is necessary to test the performance of the machine learning model and observe the prediction accuracy of machine learning through test results. *R*^2^ and *MSE* are used to evaluate the prediction accuracy of machine learning in this study, and the results of testing the machine learning model are shown in Table 4. The *R*^2^ of Ra in the training and test sets were 0.9908 and 0.9843, respectively; these two results are very close to 1 and not too far apart, indicating no overfitting or underfitting of CatBoost prediction results. The MSEs of the training and test sets were low. After evaluating CatBoost by *R*^2^ and *MSE*, it can be concluded that CatBoost can make accurate predictions for the results derived from the experiments in this study.

In the prediction process of regression models for machine learning, the prediction accuracy of the training set is usually better than that of the test set [28]. A comparison of the CatBoost prediction results with the experimental results is shown in Figure 5. Both the test and training sets of Ra are not far from the standard straight line, indicating that the predicted values of CatBoost are generally consistent with the experiments.

Different machine learning regression models have different predictive performance. In order to highlight the superiority of CatBoost prediction performance, in this study, a comparative analysis is performed by two similar machine learning models, eXtreme Gradient Boosting (XGBoost) and Adaptive Boosting (AdaBoost). The prediction results of the AdaBoost and XGBoost regression models for the input data of the training and test sets are shown in Figure 6; the prediction results of AdaBoost and XGBoost differ significantly from the experimental results.

The *MSE* and *R*^2^ of the prediction results of the three machine learning models are shown in Figure 7. The *MSE* of CatBoost is the smallest and the *R*^2^ is the closest to 1. The results indicate that CatBoost has the best prediction performance in this study.

## 5. Experimental Verification

The optimal process parameters for minimizing the surface roughness of the hemispherical CM by STF-MUM processing were solved by genetic algorithm and are shown in Table 5.

Five sets of repeated experiments were carried out according to the optimum process parameters, and the experimental results are shown in Figure 8. The surface roughness results obtained from the five sets of repeated experiments were 121.36, 123.75, 129.86, 134.89, and 139.96 nm.

A comparison of the experimental results and the optimal solution solved by the algorithm is shown in Figure 9. The values of all five sets of repeated experiments are slightly larger than the optimal solution, which indicates that the CatBoost-GA model has a high prediction accuracy.

The comparison between the experimental results and the optimal solution shows that the maximum absolute error between the five sets of repeated experiments and the optimal solution solved by CatBoost-GA is 7.21%, the average absolute error is 4.69%, and the minimum surface roughness is reduced by 28.72% compared to the minimum value of the experimental results without optimization. These results indicate that CatBoost-GA optimization is effective, and the surface roughness of the micro-hemispheric CM machined by STF-MUM is significantly reduced by CatBoost-GA optimization. Among the optimal process parameters solved by the genetic algorithm, the concentration of PPM is 10.542%, indicating that the STF plays a role in the ultrasonic processing.

## 6. Conclusions

A STF assisted MUM method was used to process hemispherical CM on silicon wafer surfaces. The box design in the response surface methodology was used for the design of experiments, and 46 experiments were carried out. The experimental results were analyzed by ANOVA to obtain the main factors affecting the surface roughness of the hemispherical CM. The ANOVA results showed that the significant influence terms included PPM concentration, abrasive concentration, ultrasonic energy, and mean particle size. The average particle size × the average particle size was the largest influence term.

In this study, CatBoost-GA model was developed for process parameter optimization, and the performance of CatBoost was evaluated by *MSE* and *R*^2^. The results show that CatBoost has better prediction performance relative to XGBoost and AdaBoost.

The optimal solution was verified by five sets of repeated experiments. The comparison between the experimental results and the optimal solution shows that the maximum absolute error between the five sets of repeated experiments and the optimal solution solved by CatBoost-GA is 7.21%, the average absolute error is 4.69%, and the minimum surface roughness is reduced by 28.72% compared to the experimental results without optimization. The results show CatBoost-GA optimization is effective. The surface roughness of micro-CM can be significantly reduced by CatBoost-GA optimization.

## Figures and Tables

**Figure 1 materials-16-02683-f001:**
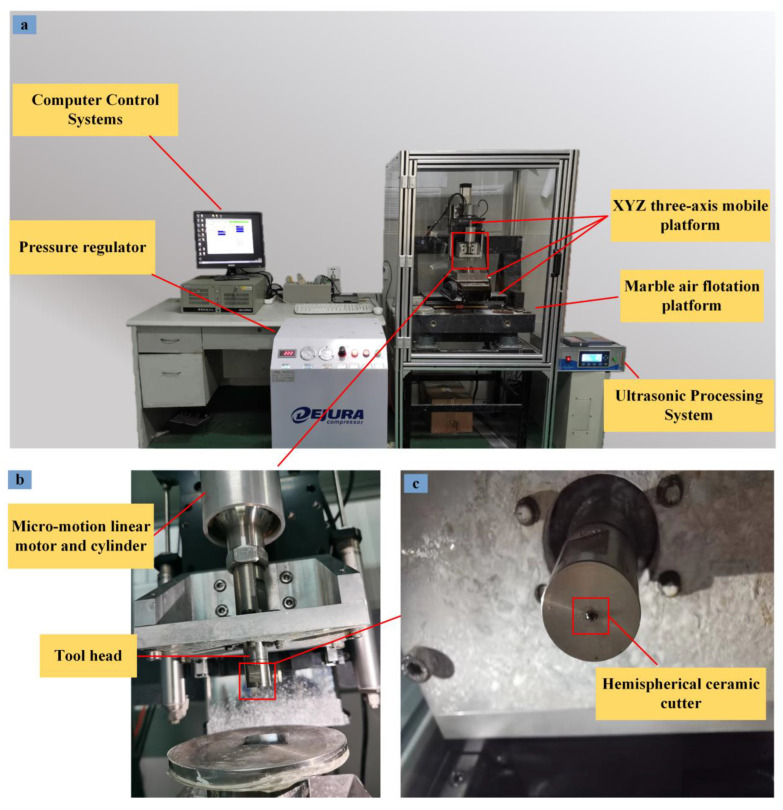
Experimental platform for machining micro-hemispheric concave molds. (**a**) Experimental platform, (**b**) z micro-axis, (**c**) tool head.

**Figure 2 materials-16-02683-f002:**
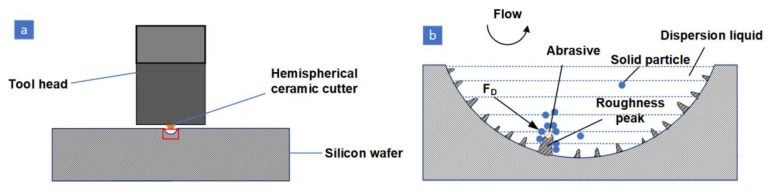
STF assisted MUM machining schematic. (**a**) Processing diagram, (**b**) processing microscopic schematic.

**Figure 3 materials-16-02683-f003:**
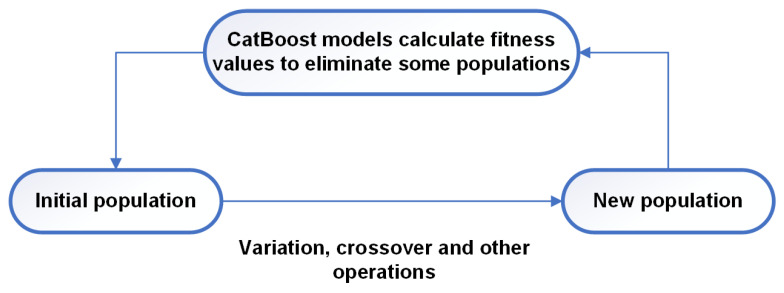
CatBoost and GA interaction diagram.

**Figure 4 materials-16-02683-f004:**
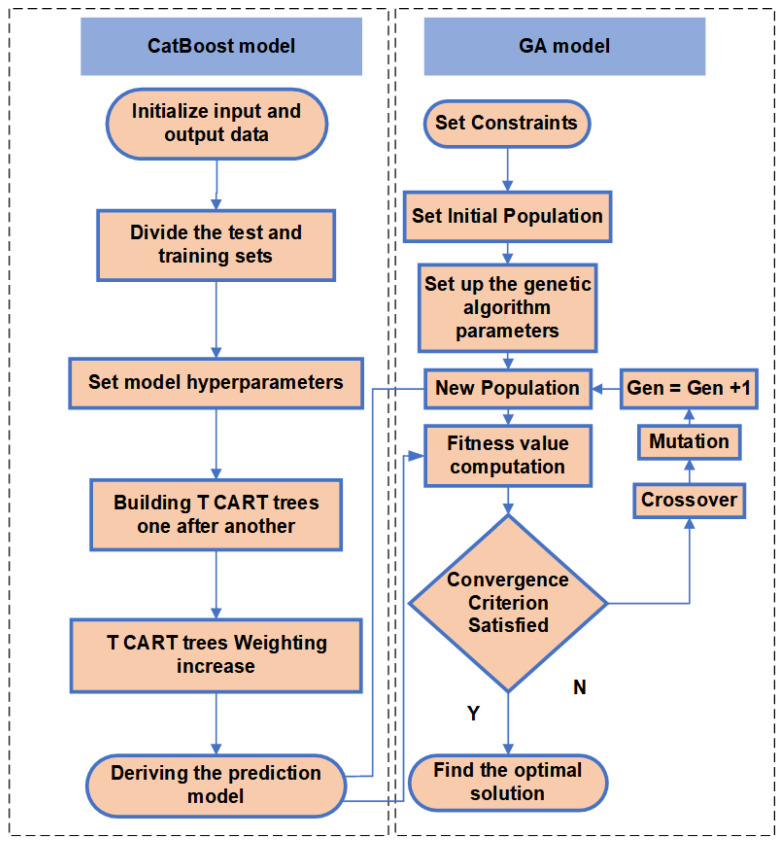
CatBoost-GA overall flow chart.

**Figure 5 materials-16-02683-f005:**
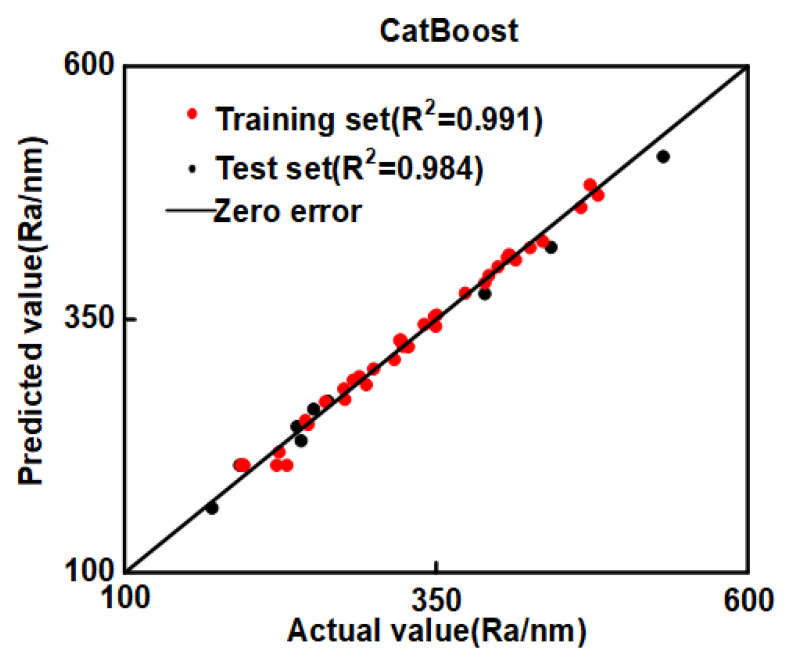
CatBoost results comparison chart.

**Figure 6 materials-16-02683-f006:**
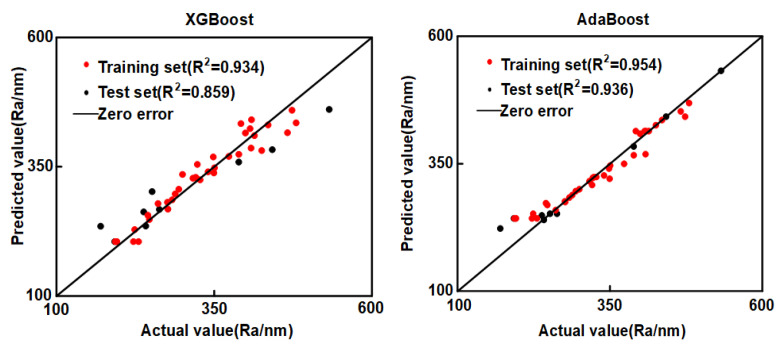
XGBoost and AdaBoost results comparison chart.

**Figure 7 materials-16-02683-f007:**
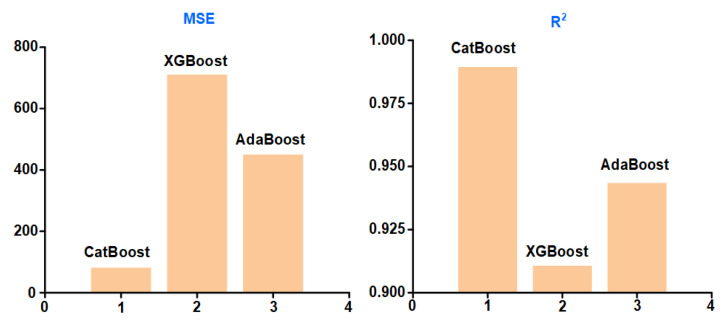
Comparison of machine learning results.

**Figure 8 materials-16-02683-f008:**
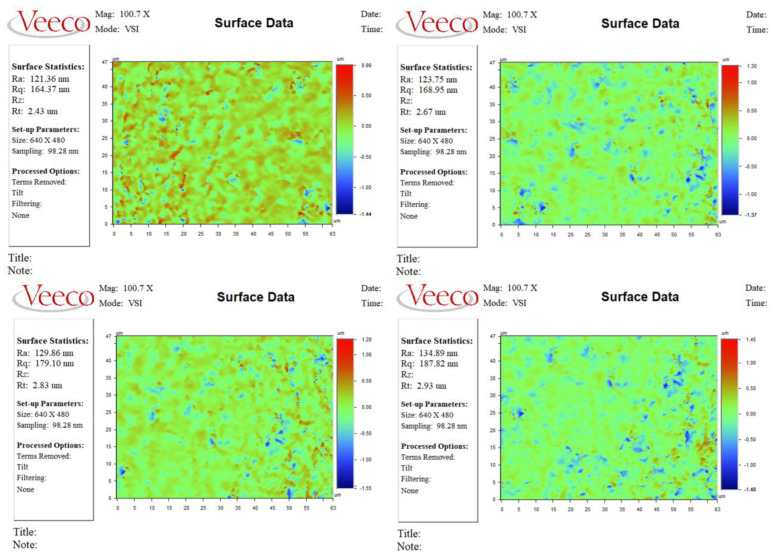
Surface roughness results obtained from five sets of repeated experiments.

**Figure 9 materials-16-02683-f009:**
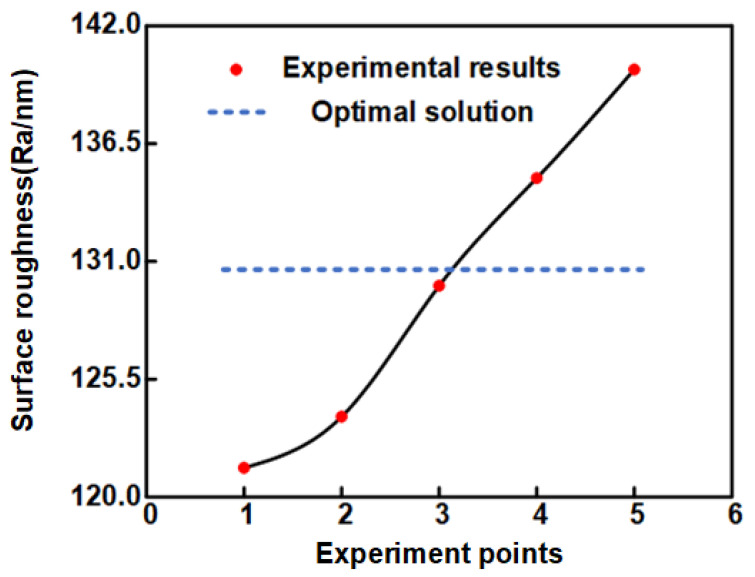
Comparison of experimental results with predicted results.

**Table 1 materials-16-02683-t001:** Process parameters and their levels.

Factors	Symbol	Levels		
		−1	0	1
PPM concentration (%)	A	5	15	25
Abrasive concentration (%)	B	4	8	12
Ultrasonic energy (w)	C	72	84	96
Tool lowering speed (μm/s)	D	1	2	3
Average particle size (μm)	E	1	2	3

**Table 2 materials-16-02683-t002:** Experimental datasets.

Run Order	A	B	C	D	E	Ra (nm)
1	−1	0	0	0	−1	170.26
2	−1	0	−1	0	0	238.42
3	−1	0	0	−1	0	241.77
4	−1	−1	0	0	0	251.76
5	−1	0	0	1	0	263.23
6	−1	0	1	0	0	389.03
7	−1	1	0	0	0	442.12
8	−1	0	0	0	1	532.22
9	0	0	0	0	0	192.44
10	0	0	0	0	0	193.4
11	0	0	0	0	0	195.47
12	0	0	0	0	0	195.85
13	0	0	0	0	0	222.08
14	0	0	0	0	0	230.24
15	0	0	−1	−1	0	224.1
16	0	0	1	1	0	245.07
17	0	0	−1	1	0	247.36
18	0	0	0	−1	−1	261.08
19	0	1	0	−1	0	276.01
20	0	0	1	−1	0	276.76
21	0	−1	0	−1	0	283.47
22	0	−1	−1	0	0	316.44
23	0	−1	0	1	0	320.75
24	0	1	−1	0	0	321.78
25	0	1	0	1	0	327.8
26	0	0	−1	0	−1	340.35
27	0	1	0	0	−1	348.61
28	0	0	0	1	−1	349.62
29	0	1	1	0	0	350.51
30	0	0	1	0	−1	373.29
31	0	−1	1	0	0	389.13
32	0	0	−1	0	1	399.55
33	0	0	1	0	1	406.95
34	0	−1	0	0	−1	408.46
35	0	−1	0	0	1	409.14
36	0	0	0	−1	1	413.81
37	0	0	0	1	1	435.55
38	0	1	0	0	1	473.43
39	1	0	0	−1	0	288.36
40	1	0	0	1	0	293.97
41	1	0	1	0	0	299.85
42	1	1	0	0	0	323.42
43	1	0	0	0	1	392.4
44	1	0	−1	0	0	425.39
45	1	−1	0	0	0	466.15
46	1	0	0	0	−1	479.83

**Table 3 materials-16-02683-t003:** ANOVA for Ra.

Factors	Degrees of Freedom	Sum of Squares	Mean Squares	F	*p*	
Model	20	350,100	17,505.52	24.65	<0.0001	significant
A (um)	1	12,129.44	12,129.44	17.08	0.0004	
B (%)	1	21.09	21.09	0.0297	0.8646	
C (w)	1	2948.06	2948.06	4.15	0.0523	
D (μm/s)	1	2969.62	2969.62	4.18	0.0515	
E (μm)	1	33,447.11	33,447.11	47.09	<0.0001	
AB	1	27,738.24	27,738.24	39.06	<0.0001	
AC	1	19,064.71	19,064.71	26.84	<0.0001	
AD	1	62.75	62.75	0.0884	0.7687	
AE	1	50,483.80	50,483.80	71.08	<0.0001	
BC	1	483.25	483.25	0.6804	0.4172	
BD	1	52.69	52.69	0.0742	0.7876	
BE	1	3853.06	3853.06	5.43	0.0282	
CD	1	754.79	754.79	1.06	0.3125	
CE	1	163.16	163.16	0.2297	0.6359	
DE	1	1115.33	1115.33	1.57	0.2217	
A^2^	1	43,217.88	43,217.88	60.85	<0.0001	
B^2^	1	67,482.73	67,482.73	95.02	<0.0001	
C^2^	1	21,110.45	21,110.45	29.72	<0.0001	
D^2^	1	546.14	546.14	0.7690	0.3889	
E^2^	1	143,900	143,900	202.58	<0.0001	
Residual	25	17,755.67	710.23			
Lack of fit	20	16,360.28	818.01	2.93	0.1182	not significant
Pure Error	5	1395.39	279.08	24.65		
Cor Total	45	367,900				

**Table 4 materials-16-02683-t004:** Evaluation metrics for CatBoost.

Dataset	Performance Metric	Surface Roughness (Ra)
Training	*R* ^2^	0.9908
	*MSE*	59.4319
Testing	*R* ^2^	0.9843
	*MSE*	178.9357

**Table 5 materials-16-02683-t005:** Table of optimum process parameters.

Run Order	A	B	C	D	E	Ra (nm)
1	10.542	6.541	81.478	0.503	0.11	130.787

## Data Availability

Not applicable.

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
