# Peer review of "Processing Optimization of Shear Thickening Fluid Assisted Micro-Ultrasonic Machining Method for Hemispherical Mold Based on Integrated CatBoost-GA Model"

_materials, 2023, doi:10.3390/ma16072683_

Round 1

Reviewer 1 Report

The paper presents micro-hemispherical concave silicon wafers processed by micro-ultrasonic micro-machining assisted by shear thickening fluids. A hybrid Categorical Boosting - genetic algorithm (CatBoost-GA) prediction model was developed and used for prediction and optimisation of surface roughness of micro-hemispherical concave moulds.

The paper title should be reworded to be shorter and more suggestive.

Pag. 3 There are 6 sections not 5. Section 1 does not show experimental equipment, experimental design, etc. Section 2 does not describe the principles, algorithmic structure, etc. These are presented in section 3. Section 3 does not show the performance of CatBoost machine learning. This is presented in section 4. In Section 4, the optimal prediction solution of the CatBoost-GA model is not verified experimentally. This is the subject of section 5.

There are inconsistencies / between what the text says and Table 1 (e.g. in the text it is mentioned D represents abrasive particle size, but in table 1, D is the Tool lowering speed). Although these are presented as the main influencing factors, in the conclusions it is stated that the PPM concentration had a significant effect (without mentioning to what extent) and the significance of the other factors is not stated.

Table 2 with experimental data sets must be review since it contain errors. Only 5 central points are presented: run order 9 to 13 (not 6 points as was mentioned in the text).

Text related to figure 3. Only diagrams are presented, not the supporting mathematical models. It is not clear the algorithm of elimination of some populations. The mathematical expressions underlying the algorithm are not presented. Recommendation: If it's something new, it should be fully explained. If not, you don't need to show all the diagrams (you can only indicate the reference to a bibliography; the article should not be unnecessarily long).

From the study of Figure 7 when using different XGBoost and AdaBoost prediction methods, different dispersions of the training set values are found. It should be explained why this happens. Isn't it the same training set of data /actual experimental data used? Also explain why the test set always has a higher dispersion than the training set (even though the training set is based on more data).

In figure 10 in the case of the experimental results was not plotted between the points but joined the points. Why? The roughness on the y-axis should be mentioned.

Editorial errors:
Have to be used μm (not um) - in the entire document
output vector are repeated (pag.7);  X is input vector
table 3: pure error (not errot)
review the eq. 4 (using ";" instead of "," to not create confusion)
Different font sizes are used, needs to be corrected (e.g. conclusion section)
There are certainly more errors, the paper was treated superficially.

English needs to be revised.

Figures 9 do not have the appropriate resolution.

In the conclusions it is stated that PPM concentration had a significant effect on the experimental results. Nothing is said about the other significant factors (e.g. the average particle size, ultrasonic energy).

Author Response

Dear reviewer,Please see the attachment.

Reviewer 2 Report

This work offers a comprehensive and thorough understanding of the formation the importance of reducing the surface roughness of silicon micro-hemisphere concave molds inside MEMS hemispherical resonant gyroscopes for improved performance. The authors propose a combined approach (STF-MUM) utilizing the ultra-precision polishing of Shear thickening fluids (STF) and a versatile Micro-ultrasonic machining (MUM) technique to improve surface roughness. They explored the CatBoost-genetic algorithm (GA) optimization model to optimize processing parameters and developed a CatBoost-GA prediction model for target prediction and optimization of surface roughness for micro-hemispherical CM. The findings indicate that PPM concentration is an important influencing factor in STF assisted MUM, and CatBoost-GA optimization is effective in significantly reducing the surface roughness of micro-CM. The clarity and organization of the author’s presentation in this article make their insights easily accessible to researchers who are new to the field. The article is written with clarity, making it an enjoyable read, and the scientific discussion is straightforward to follow. This paper could prove to be an invaluable resource for a wide range of researchers, including those working in the areas of Micro-electro-mechanical systems, Materials technology, Ultra-precision machining, and aerospace applications.

To further enhance the article and provide additional clarity, the following suggestions are offered:

·        The authors are advised to specify the diameter of the micro hemispherical concave mold fabricated in this study.

·        As per the tool diameter of 0.8 mm, the resultant micro hemisphere CM might be ~1.5 to 2 times the diameter of the tool, depending on the depth. This leads to ~1.2mm to 1.6mm diameter of CM. Is there any specific reason for taking a very small sampling size (63x47 um) for surface roughness measurement of a millimeter order diameter CM? Kindly justify it. Has any non-uniformity in surface roughness at the bottom and side walls of CM been observed?

·        It would be helpful if the authors could provide a comparison of their surface roughness results with those reported in the literature, highlighting any improvements they achieved.

·        It would be interesting to know what the smoothest surface of any hemispherical concave molds achieved with any other techniques is, to put the author's results in context.

·        To provide more information on the practical aspects of this process, it would be useful to include a discussion of its cost and time effectiveness compared to other methods.

·      #  On page 2, the first paragraph needs to be rewritten for clarity. This paragraph also contains a typo: "During the micro-ultrasonic ………. of the material surface [14]

Author Response

Dear reviewer,Please see the attachment

Round 2

Reviewer 1 Report

The authors have made the suggested modifications/completions.